# Genetic *Listeria monocytogenes* Types in the Pork Processing Plant Environment: From Occasional Introduction to Plausible Persistence in Harborage Sites

**DOI:** 10.3390/pathogens10060717

**Published:** 2021-06-07

**Authors:** Niels Demaître, Geertrui Rasschaert, Lieven De Zutter, Annemie Geeraerd, Koen De Reu

**Affiliations:** 1Technology and Food Science Unit, Flanders Research Institute for Agriculture, Fisheries and Food (ILVO), Brusselsesteenweg 370, 9090 Melle, Belgium; niels.demaitre@ilvo.vlaanderen.be (N.D.); geertrui.rasschaert@ilvo.vlaanderen.be (G.R.); 2Department of Veterinary Public Health and Food Safety, Faculty of Veterinary Medicine, Ghent University, Salisburylaan 133, 9820 Merelbeke, Belgium; Lieven.DeZutter@UGent.be; 3Division MeBioS, Sustainability in the Agri-Food Chain Group, BIOSYST Department, KU Leuven, Willem de Croylaan 42, Box 2428, 3001 Leuven, Belgium; annemie.geeraerd@kuleuven.be

**Keywords:** *Listeria monocytogenes*, cutting plant, environment, persistence, sampling plan, after C&D, during production

## Abstract

The purpose of this study was to investigate the *L. monocytogenes* occurrence and genetic diversity in three Belgian pork cutting plants. We specifically aim to identify harborage sites and niche locations where this pathogen might occur. A total of 868 samples were taken from a large diversity of food and non-food contact surfaces after cleaning and disinfection (C&D) and during processing. A total of 13% (110/868) of environmental samples tested positive for *L. monocytogenes*. When looking in more detail, zone 3 non-food contact surfaces were contaminated more often (26%; 72/278) at typical harborage sites, such as floors, drains, and cleaning materials. Food contact surfaces (zone 1) were less frequently contaminated (6%; 25/436), also after C&D. PFGE analysis exhibited low genetic heterogeneity, revealing 11 assigned clonal complexes (CC), four of which (CC8, CC9, CC31, and CC121) were predominant and widespread. Our data suggest (i) the occasional introduction and repeated contamination and/or (ii) the establishment of some persistent meat-adapted clones in all cutting plants. Further, we highlight the importance of well-designed extensive sampling programs combined with genetic characterization to help these facilities take corrective actions to prevent transfer of this pathogen from the environment to the meat.

## 1. Introduction

*Listeria monocytogenes* is an ongoing pathogen of concern regarding food safety, particularly in ready-to-eat (RTE) foods, including RTE meat products. Pork and processed pork products have been implicated in outbreaks across Europe, with important human suffering from disease symptoms as well as significant economic repercussions resulting from recalls, irreparable image-damage, and even closures of food production facilities [1,2].

The persistence of *L. monocytogenes* in the processing environment is considered to be the primary source of food contamination [2]. This pathogen has established a reputation as a common colonizer of food processing environments and thus represents a contamination risk [3]. Moreover, foodborne listeriosis outbreaks have been linked to environmental *L. monocytogenes* contamination at processing companies [4]. This highlights the importance of preventing its introduction and persistence in food processing environments. In order to prevent the introduction and establishment of the pathogen in meat processing companies further down the agri-food chain, control of this pathogen at primary production, at the slaughterhouse, and cutting plant level is critical. Studies showed contamination by this pathogen in pork can originate on the farm [5]. We also recently confirmed that some Belgian pig slaughterhouses are still experiencing a significant presence of *L. monocytogenes* on carcasses at the end of the slaughter line, which may possibly be responsible for initial settlement of specific strains in the post-slaughter processing environment [6]. Environmental contamination in chilling and cutting areas is a significant cause of meat contamination with *L. monocytogenes* [7,8,9].

The ongoing risk of reintroduction and cross-contamination of pork via the environment and equipment (and vice versa) hinders efforts to detect and eliminate this pathogen. Meat processing plants are under tremendous pressure, as Whole Genome Sequencing (WGS) -based typing makes it possible to link outbreak strains to a common source such as one company. For these reasons, the industry is pursuing improved and science-based environmental monitoring strategies to identify contamination sources and take corrective action.

The objectives of the current study were (i) to investigate the *L. monocytogenes* occurrence and genetic diversity in Belgian pork cutting plant environments and (ii) to identify harborage sites and niche locations where *L. monocytogenes* might survive.

## 2. Results

### 2.1. Contaminated Surfaces

#### 2.1.1. General Results

A total of 868 surface samples were taken in three pork processing plants. Of these, 110 [13%; 95% CI: 10–15%] tested positive for *L. monocytogenes*. A clear difference in the number of *L. monocytogenes* positive samples between surfaces in the three contaminated zones was observed: zone 1 had 6% positive (25/436), zone 2 had 8% (13/154), and zone 3 had 26% (72/278), aggregating results both after C&D and during production.

No difference was observed in the number of positive samples after C&D (13%; 57/428) and during processing (12%; 53/440). These differences were also limited when assorting by zone. After C&D and during production, zone 1 had 5% and 7%, zone 2 9% and 8%, and zone 3, 28% and 24%, respectively.

The overall spread of the pathogen in the processing environment is shown in Figure 1.

The highest occurrence was found on the pallet truck wheels (zone 3), both after C&D (100%; 6/6) and during production (83%; 5/6). Other non-food contact surfaces (zone 3) such as plastic transport pallets, floors, floor drains, and cleaning equipment were also frequently contaminated, ranging from 44% to 83%. Zone 2 surfaces, including handles of circular saws and cutting boards and the undercarriage of conveyor belts, were occasionally *L. monocytogenes* positive. After C&D, the presence was limited to a few food contact surfaces (zone 1), such as circular saws, meat cutting tables and board, hooks, conveyor belts and the inside of containers. During production, it was seen that this pathogen’s presence was more widely distributed on direct contacts. For example, *L. monocytogenes* was detected on chopping blocks, pork peeling machines, a carcass divider, a stainless steel cart, a chainmail glove, and a meat spike, but no longer detected after C&D.

#### 2.1.2. Results at Plant Level

The *L. monocytogenes* prevalence in meat processing plant A was the highest, with almost 17% (48/291), followed by 13% (35/277) in plant B. Plant C had the lowest prevalence, 9% (27/300). An overview of the percentage of positive surface samples at the meat processing plant level and per sampling day is shown in Table 1.

### 2.2. Molecular Typing

#### 2.2.1. Genetic Diversity

In total, 110 *L. monocytogenes* isolates were collected from cutting plant environments. Multiplex PCR-based serotyping resulted in two predominant serogroups: IIa and IIc, which represented 66% and 33% of the isolates, respectively. Serogroup IIb was rare, representing only one isolate (1%). PFGE results showed a low genetic diversity within the *L. monocytogenes* isolates. With a delineation level of 85%, only 11 pulsotypes were obtained, correlating to 11 clonal complexes (CC): CC9, CC31, CC8, CC121, CC29, CC11, CC20, CC37, CC89, CC155, CC5 (Figure 2). Clearly, the first four (CC9, CC31, CC8, and CC121) were most prevalent, representing 93% (102/110) of all isolates.

Most of these common CCs were isolated at two different sampling times in multiple meat processing plants (Figure 3). CC9 was the most abundant complex (36/110; 33%) isolated in all three plants. It was isolated 12, 9, and 15 times in plants A, B, and C, respectively during two sampling days. Within CC9, two subgroups could be distinguished, CC9-P1 (*n* = 35) and CC9-P2 (*n* = 1). The second most common clone, CC31 (33/110; 30%), was found in two plants during two sampling events. In plant A, this clone was isolated 17 times and in plant B 16 times. Within this cluster, two unique pulsotypes could be differentiated, namely, CC31-P1 (*n* = 26) and CC31-P2 (*n* = 7); CC31-P2 was only found on the first sampling day in plant A. CC8 was mainly isolated in plant A, where it was recovered 17 times, and to a lesser extent, two and five times in plants B and C, respectively. Three pulsotypes were found within CC8. CC8-P1 was only isolated in cutting plant A. CC8-2 was found in cutting plant C and once in cutting plant B, where CC8-3 was also recovered once. CC121 was isolated seven times in plant B across two sampling days and only once in plant A.

#### 2.2.2. Distribution of Clonal Complexes in the Environment

A detailed overview of the appearance and genetic diversity of the *L. monocytogenes* isolates is also shown in Figure 3, classified by environmental sampling zones, both after C&D and during processing for each cutting plant and per sampling day. Results showed the widespread prevalence of common *L. monocytogenes* complexes and identical pulsotypes in all cutting plant environments over time.

CC9-P1 was mainly found on zone 3 surfaces in all meat processing plants and after C&D sporadically in zone 1 and 2 on a circular saw, the conveyor, and meat hooks. CC31 was isolated in meat processing plants A and B. In plant A, both CC31-P1 and CC31-P2 were found. In plant B, only CC31-P1 was found in zones 1 and 3. CC8-P1 was predominant in plant A (71%; 17/24) where it was found on surfaces from all three zones. Notably, during the second visit, this pulsotype was primarily isolated from zone 1 surfaces during pork cutting, including the cutting table and conveyor belt. Two CC8 isolates were recovered in plant B, one from the exterior of meat containers (CC8-P2) and one from the pork skin peeling machine (CC8-P3). In plant C, CC8-P2 was only found on zone 3 surfaces, mainly after C&D. CC121-P1 was isolated mainly in plant B on the floor, the pallet truck (zone 3), and a meat container (zone 1).

## 3. Discussion

In this study, the prevalence and genetic diversity of *L. monocytogenes* in three Belgian pork meat cutting plants was investigated via extensive environmental sampling. Results suggested that (i) cutting plants are colonized over time by *L. monocytogenes*, predominantly in zone 3; (ii) a small group of meat-adapted clonal complexes is widely distributed and may persist in cutting plant environments, mainly at harborage sites; and (iii) that well-designed sampling programs performed over time and combined with genetic characterization are important to distinguish reintroduction and persistence. This study provides insights into harborage sites and niche locations and consequently possible meat contamination from the environment.

### 3.1. Meat Processing Facilities Are Expected to Be Colonized by L. monocytogenes

In total, 13% of the environmental samples tested positive for *L. monocytogenes*. All meat processing plants were found positive during the sampling visits, with an overall *L. monocytogenes* prevalence of 12% after C&D and 13% during processing. These results appear low compared to a Canadian study, where nearly 42% of samples collected in one pork processing plant after C&D were positive [10]. Environment and equipment samples taken at dry-cured ham processing facilities in Spain were contaminated at similar rates to our study after C&D (11%) and were contaminated at higher levels during processing (25%) compared to our study [11]. In French dried sausage processing plants, the proportions were 15% before the beginning of the working day and 47% during processing [12]. In a newly opened meat processing facility, *L. monocytogenes* was absent in the environment on the first day of production. In contrast, after several months of production, a persistent clone had already colonized the facility with a prevalence of up to 77% of surfaces in zone 1, indicating the rapid colonization potential of this pathogen as soon as contaminated raw materials were introduced [13]. In our study, the plant with the most modern design and newest infrastructure showed the lowest number of surfaces testing positive in comparison to the other older plants, although these differences were quite limited. The above findings show that *L. monocytogenes* contamination appears to be inevitable in pork processing environments, highlighting the importance of environmental monitoring at cutting and subsequent stages to prevent pathogen transfer from the environment [2,10].

### 3.2. Food Contact Surfaces Are Contaminated, Even after C&D

In the present study, food contact surfaces (zone 1), which pose an immediate risk for meat contamination, were occasionally contaminated. Even after C&D, surfaces such as circular saw blades, meat hooks, cutting tables/boards, the interior surfaces of meat containers, and conveyor belts remained sporadically contaminated. This indicates a need for corrective action and staff training to increase awareness and prevent future contamination and persistence on these surfaces. The worktop of cutting tables and cutting boards often showed extensive cutting damage, which hinders effective cleaning and disinfection. More complex, hard-to-clean processing machines such as circular saws are susceptible to persistent *L. monocytogenes* contamination and make it difficult to completely eliminate the pathogen [14]. However, pork skin peeling and de-rinding machines, despite their complex machinery and hard-to-clean nature, were not found to be contaminated after C&D, possibly due to increased awareness and care by cleaning personnel. It is also possible that the pathogen may have gone undetected as pork skin peeling and de-rinding machines were either not operated or were only operated for a short time when taking samples after C&D, leaving the organism in the harborage sites at that time [4,15,16].

During production, food contact surfaces such as chainmail gloves, cutting tables/boards, conveyor belts, peeling machine, carcass divider, and chopping blocks tested positive for *L. monocytogenes*, suggesting the potential for repeated contamination of cuts of meat. Insufficient C&D of food contact surfaces before production or cross-contamination from non-food contact surfaces and incoming raw materials are also possible causes.

### 3.3. Most Frequently Contaminated Zone 3 Surfaces Are Typical Harborage Sites in Processing Environments

Non-food contact surfaces located some distance away from exposed products but still within the exposed product area (zone 3) were more likely to be contaminated with *L. monocytogenes*, namely 65% (72/110) of the positive samples. This is consistent with other studies where non-food contact surfaces were significantly more contaminated than food contact surfaces [11,17]. The sampling of non-food contact surfaces has proven to be helpful in obtaining early indications of contamination and persistence. Drains and floors were often contaminated in our study, and are generally considered to be reliable sampling sites for monitoring *L. monocytogenes* colonization patterns [18]. These collection sites for all of the facility’s water and organic residues might become reservoirs for persistent *L. monocytogenes* and eventually contamination hotspots [15,19]. Previous studies have shown the significance of floors, drains, and other non-food contact surfaces as spreading pathways of *L. monocytogenes* in food processing facilities [18,20,21]. Floor cleaning equipment regularly contained *L. monocytogenes* in all cutting plants, especially brushes and squeegees, which were still contaminated in 50% of the cases, even after having been cleaned for use the next working day. These data are consistent with studies conducted in other processing environments showing that drains, floors, and cleaning equipment used to maintain these sites are the main sites of contamination and may be considered the main harborage sites for this pathogen [17]. Furthermore, pallet trucks, particularly the wheels and plastic pallets, were also found to be niche locations for *L. monocytogenes* in our study. This should be considered when operating pallet trucks as the pathogen may be transferred from one production area to another.

After C&D, zone 3 surfaces were often (and in many cases even more) contaminated than during production, even though detection may be complicated by the use of chemical agents that damage living cells and thus render them non-culturable [11]. The neutralization of the disinfectant’s residual action on microbiological growth by the Dey-Engley broth used during sampling might have increased the detection probability.

### 3.4. Low Genetic Heterogeneity and the Widespread Prevalence of Meat-Adapted Clones

The vast majority of *L. monocytogenes* isolates from cutting plant environments appear to belong to serogroups IIa and IIc, except for one isolate that belonged to serogroup IIb. This is in accordance with other studies reporting the more common presence of these serogroups along with serogroup IIb in pork meat plants [9,22,23]. PFGE analysis showed a low genetic heterogeneity. Notably, four CCs (CC8, CC9, CC31, and CC121) dominated all zone surfaces both after C&D and during production.

CC121, CC8, CC9 are the most common genotypes in pork processing environments [24]. A study mentioned that CC8 strains had a strong biofilm formation potential, which might support persistence within food production environments [25]. Also, CC8 clones were associated with human listeriosis cases and outbreaks across Canada for more than two decades [26]. Several studies reported the favorable settlement and the persistence of CC9 in meat processing environments and equipment for several years [6,24,27,28,29,30]. This clone possesses higher stress resistance and benzalkonium chloride tolerance genes and a biofilm formation ability that contributes to its persistence [29,30]. Likewise, CC31 strains showed a higher frequency in meat and meat products [2,28].

These findings show a better adaptation of these complexes to raw pork and pork processing environments, suggesting (i) the continuous introduction and repeated contamination via incoming carcasses due to a higher representation of these complexes and/or (ii) their persistence in the concerned cutting plants, probably by genetic determinants contributing to their establishment.

### 3.5. Continual Reintroduction (i) or Actual Persistence (ii)

Contamination may initially originate from CCs at the farm [5]. Certain strains may be more widely distributed in nature, thus being more easily introduced and reintroduced to processing plants via raw material [31]. However, this is not apparent from our results, as the dominant CCs in this study were not or hardly found on farm levels, except for CC8 [24]. It is also possible that these dominant CCs might settle in the preceding step in the meat chain, namely in slaughterhouse environments, through which a continuous introduction and eventual contamination might occur via incoming carcasses. Our previous study showed the regular contamination of the ventral and anterior pig carcass sites with CCs 21, 37, and 224 [6]. However, these carcass contaminations were more linked to contamination during slaughter, originating from incoming pigs than from environmental contamination through persistent well-established complexes. Nevertheless, the often contaminated pig carcasses pose significant contamination risks due to their repeated (re-) introduction into cutting plant environments and the possible establishment of specific complexes. Thus, it has already been proven that *L. monocytogenes* isolates in meat processing units originated from the slaughter line [32].

More plausible is that the unique pulsotypes were persistently present in all meat processing plants, as they were widespread and found repeatedly, especially in zone 3. Persistent contamination poses high risks of spreading this pathogen across processing facilities, which could explain the ubiquitous presence of these pulsotypes both after C&D and during production. Contamination from persistence is more likely to occur after routine cleaning and sanitizing have become ineffective as, compared to less commonly isolated complexes introduced sporadically and presumably considered transient.

### 3.6. Evaluation of Persistence Demonstrated by PFGE

As well-described by Carpentier and Cerf [33], persistence is a loosely defined concept generally referring to the repeated isolation of identical strains ascertained by the same molecular technique over time. In particular, the number of isolation events and the period during which an identical strain is isolated vary depending on the authors. According to Lunden et al. [14], pulsotypes isolated at least five times over three months from meat and poultry processing plants were considered persistent. Therefore, in our study, the number of sampling moments and time between sampling days proved insufficient to make a statement about persistence according to this definition. Still, the repeated widespread prevalence revealed in this study of unique pulsotypes from surfaces at cutting plant level both after C&D and during production, combined with the current knowledge about the assigned clonal complexes, more specifically CC8, CC9, CC31, and CC121, clearly indicate the likelihood of persistence. This is certainly the case when our approach to a more in-depth delineation of AscI and ApaI band patterns is considered, showing a deeper classification of visually completely identical pulsotypes. Additionally, over-discrimination of single *L. monocytogenes* clones can occur because intact prophage regions result in PFGE profiles differing by up to three bands, leading to persistence being overlooked [34].

WGS-based typing, which has higher discriminative power than PFGE analysis, should further divide the genetic groups at the plant level, although PFGE apparently could differentiate *L. monocytogenes* isolates beyond the CT level [35]. The question arises as to what extent isolates should be characterized to determine persistence and provide support at the meat processing plant level.

More research is needed to reveal whether the pulsotypes found across different plants are plant-specific or not. A study by Autio et al. [31] did show the repeated isolation of the same pulsotypes from pork products in different unrelated plants. Still, WGS improved discriminatory power over PFGE by differentiating isolates with identical pulsotypes obtained from various delis into plant-specific genotypes [34].

In conclusion, unique pulsotypes and even WGS-based classification typically do not necessarily provide enough information to determine whether re-isolation of a specific complex, in particular, is either transient or actual persistent in meat processing plant environments [34]. Still, PFGE and MLST can be used as robust typing techniques while transitioning towards WGS. Primarily well-designed risk-based proactive sampling programs that involve consequent and frequent sample collection after C&D and during production combined with genetic characterization will remain crucial to distinguishing reintroduction and persistence. A constant revision of these plant-specific sampling programs should be considered based on historical data where the feasibility and the additional costs involved, regardless of the typing technique, might be the main obstacle.

## 4. Materials and Methods

### 4.1. Meat Processing Plant Selection and Description

Three pork processing plants located in Flanders (the northern region of Belgium) were included in the study. All plants represented small-sized companies with 25 to 35 employees. Plants A (100–200 pigs per day) and B (800–900 pigs per day) are older, and pig carcasses are delivered in both companies by the proprietary slaughterhouse located at the same sites. Plant C is a company (500 pigs per day) with 30 employees and has a new building, modern design, and entirely new infrastructure complying with the most stringent standards and quality demands. Local slaughterhouses supply pig carcasses.

Each plant was visited twice from November 2019 to July 2020. The time elapsed between the two sampling events varied from 5 to 20 weeks. Originally the intention was to sample each plant twice with an interval of 4–6 weeks to acquire a thoroughgoing overview of the situation over a month. Due to the COVID 19 outbreak, sampling at plant A had to be postponed bringing the time between two sampling events to 20 weeks. All three plants had implemented HACCP principles and GMP, according to the European legislation. Alkaline chlorine-based products were the most commonly used sanitizers in the plants visited.

### 4.2. Environmental Sampling

Intensive and detailed sampling of specific locations and equipment was performed. Between 128 and 150 environmental and equipment samples from processing workrooms and cooling rooms were collected to detect *L. monocytogenes* per sampling day. The number of samples taken from surfaces varied due to infrastructural differences between the three plants and specific equipment availability at the different sampling times. A comparable number of samples was collected, during both sampling days together, from each meat processing plant, namely 291, 277, and 300 in plants A, B, and C, respectively. Overall, a total of 868 surface samples were tested for the presence of *L. monocytogenes*.

Determination of which surfaces were sampled was performed in consultation with staff of the plants, with attention to critical sampling locations likely to harbor *L. monocytogenes* [36]. During each sampling occasion, sampling was performed after cleaning and disinfection (C&D), as well as during production. First, sampling was performed immediately after C&D operations, where the sampling of specific mobile electrical equipment (circular saws, pork skin peeling, and de-rinding machines) was performed, if possible, following a brief run of the device to loosen possible dirt and pathogens. Secondly, sampling was performed during processing, after 4 ± 2 h of production, to possibly increase the probability of *L. monocytogenes* detection from biofilms and niches [16].

The USDA FSIS zone classification for environmental sampling was used with a risk level attributed to the sampled areas according to the risk of exposure to the food products [4]. This classification system describes four sampling zones with decreasing risk levels for food contamination. The three highest risk zones were applied in this study’s sampling plan: (i) zone 1 is the highest risk area, consisting of direct food contact surfaces; (ii) zone 2 contains indirect contact surfaces that are physically close to the food product but not in contact with the product (i.e., the undercarriage of conveyor, machine handle or frame); (iii) zone 3 contains surfaces away from exposed product but which are still in the exposed product area (i.e., floors, drains, and undersides of equipment) [4]. A detailed overview of the sampled surfaces per zone type and the range of the number of samples taken from a given surface during one sampling day (N) is shown in Figure 1.

Samples of surfaces taken after cleaning and disinfection were collected using sponge-sticks (3M, St. Paul, MN, USA) pre-moistened with 20 mL of sterile Dey-Engley neutralizing broth (Sigma Aldrich, St. Louis, MO, USA). Sponge-sticks for samples taken during processing were pre-moistened with 20 mL of sterile Maximum Recovery Diluent (MRD; Oxoid, Basingstoke, UK). Before swabbing, the absorbed liquid was squeezed from each sponge into the bag to prevent fluid loss. Hard-to-access sites were sampled using a combination of cotton swab sticks (Cultiplast^®^ swab, LP Italiana, Milan, Italy) pre-moistened with 2 mL of sterile Dey-Engley neutralizing broth or MRD and sponge-sticks; both surface sampling types were pooled. Sampled surface areas varied between 10 cm² and 1000 cm² depending on the surface type. All samples were transported under cooled conditions to the laboratory, where they were kept at 3 ± 2 °C and processed the same day.

### 4.3. Microbiological Analyses

Detection of *L. monocytogenes* was based on ISO11290-1:2017. Initial volumes of Dey-Engley neutralizing broth or Maximum Recovery Diluent were enriched after adding 20 mL of double concentrated Fraser broth to obtain a half concentrated Fraser solution. Before further analysis, the enriched swab samples were homogenized for 2 min using a Stomacher Lab Blender 400 (Seward Laboratory, London, United Kingdom). After incubating at 30 °C for 24 h, 0.1 mL was transferred into 10 mL Fraser broth tubes (Bio-Rad, Marnes-La-Coquette, France) and incubated at 37 °C for 24 h. Isolation was done by streaking a loopful (10 µL) of the incubated Fraser broth on Agar *Listeria* plates according to Ottaviani and Agosti (ALOA; Bio-Rad, Marnes-La-Coquette, France). Plates were incubated at 37 °C for 24 h. From each plate, one suspect *L. monocytogenes* colony was picked and further purified on ALOA plates. This isolate was then retained for further analysis. All isolates were then streaked on Tryptone Soya Yeast Extract Agar (TSYEA) plates (TSA; Oxoid CM0131, Basingstoke, UK/YE [0.6%]; Oxoid LP0021, Basingstoke, UK) and incubated for 24 h at 37 °C. All isolates were stored in 15% glycerol stocks at −80 °C for further testing.

### 4.4. Molecular Analysis

Suspected *L. monocytogenes* isolates were confirmed by multiplex PCR [37] and serotyping [38], then confirmed strains were Pulsed-Field Gel Electrophoresis (PFGE) typed. PFGE was performed according to the PulseNet standardized procedures [39] with AscI and ApaI enzymes (New England BioLabs, Ipswich, MA, USA). Similarities between AscI and ApaI fingerprint patterns were studied using BioNumerics version 7.6 software package (Applied Maths, Sint-Martens-Latem, Belgium) to assign pulsotypes from which Clonal Complex (CC) information can subsequently be derived [24,40]. The similarities between the fingerprints were calculated using the band-based Dice coefficient with an optimization and position tolerance of 1%. The clustering of fingerprints was performed using the Unweighted Pair Group Method with Average Linkages (UPGMA).

First, pulsotypes were defined when PFGE fingerprints of a group of isolates showed a similarity of more than 85%, and CCs were assigned according to the mapping protocol described in the Félix et al. study [24]. The authors kindly provided the typing data, both PFGE and MLST, from 396 isolates to deduce the CCs for the PFGE profiles within our dataset using BioNumerics. A PFGE cluster was assigned to the same CC if the profiles matched at least 85% with profiles of strains previously typed by MLST. As a result, mapping our PFGE results with MLST results was possible with a high congruence value [41]. In cases where it was not possible to assign a CC from our panel, conventional Multilocus sequence typing (MLST) was performed according to the protocol of Institut Pasteur (https://bigsdb.pasteur.fr/listeria/primers_used.html, accessed on 7 June 2021). The assignment of the sequences to the MLST types was performed with the MLST plugin of Bionumerics according to the MLST scheme of Institut Pasteur.

Second, a more in-depth delineation of AscI and ApaI band patterns within the CCs was examined, with pulsotypes considered unique when visually completely identical (e.g., CC9-P1, CC9-P2, etc.).

## 5. Conclusions

Cutting plants may be expected to be colonized by *L. monocytogenes* because of continuous reintroduction and/or actual persistence in the environment. This entails possible settlement of meat-adapted complexes, especially at niche locations, with eventual contamination at higher-risk surfaces (zone 1). Even after C&D, the pathogen was still frequently detected, especially at typical harborage sites in zone 3 and even, to a limited extent, in zone 1. We recognize that the requirements previously described to consider persistence were insufficient in this study. However, the pronounced representation of deeper classified and visually completely identical pulsotypes, combined with the knowledge about the assigned meat-adapted CCs, suggests persistence in all cutting plants. It begs the question as to how persistence is assessed and, consequently, how deep isolates should be characterized with the rise of WGS. Nevertheless, this study’s approach provides clear insights for support and highlights the importance of well-designed proactive sampling programs combined with genetic characterization to differentiate reintroduction from persistence.

## Figures and Tables

**Figure 1 pathogens-10-00717-f001:**
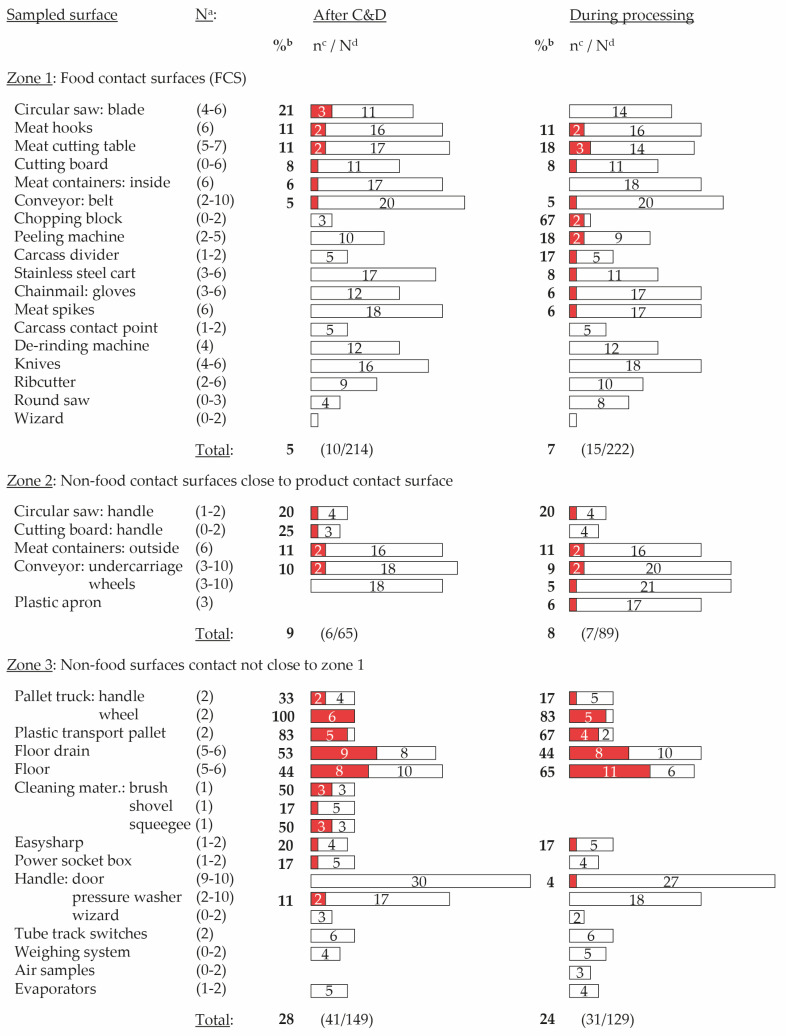
Total overview of sampled surfaces categorized according to the environmental sampling zones. ^a^ Number of samples taken each sampling day; ^b^ Percentage positive samples; Horizontal stacked bars show: ^c^ the number of *L. monocytogenes* positive samples (red) and ^d^ the number of *L. monocytogenes* negative samples (white) both after C&D and during processing.

**Figure 2 pathogens-10-00717-f002:**
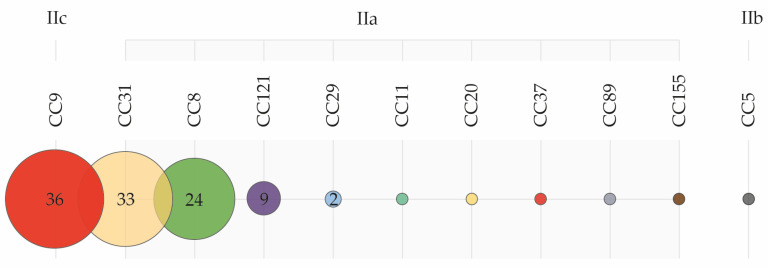
Genetic diversity of 110 *L. monocytogenes* isolates based on serogroup, and assigned clonal complexes (CC).

**Figure 3 pathogens-10-00717-f003:**
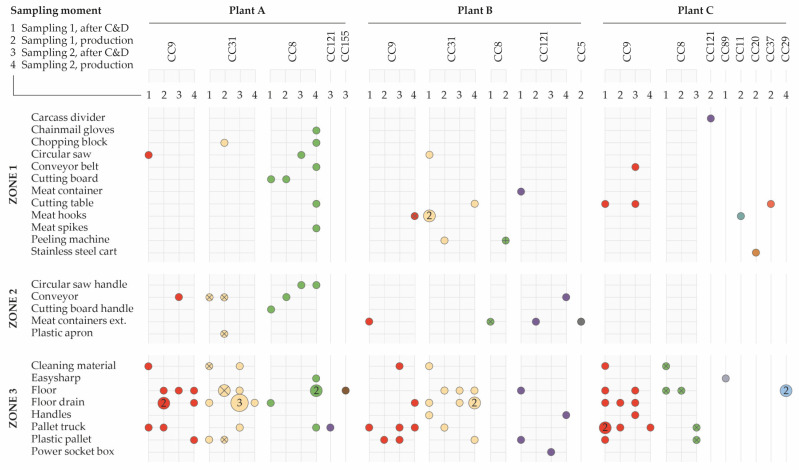
Genetic distribution of *L. monocytogenes* isolates in the environment given per cutting plant and sampling event. The colors indicate the different clonal complexes (CC), and the marking indicates unique pulsotypes within a CC (P1: ◯; P2: ⊗; P3: ⊕).

**Table 1 pathogens-10-00717-t001:** Percentage of *L. monocytogenes* positive samples given per meat processing plant and per sampling zone and day.

		Plant A				Plant B				Plant C					
Zone	Day	After C&D		During Production		After C&D		During Production		After C&D		During Production		Totals	
1	1	5%	(2/38)	6%	(2/35)	11%	(4/36)	5%	(2/41)	3%	(1/36)	11%	(4/37)	7%	(15/223)
	2	3%	(1/38)	14%	(5/35)	0%	(0/30)	5%	(2/37)	6%	(2/36)	0%	(0/37)	5%	(10/213)
2	1	17%	(2/12)	19%	(3/16)	33%	(2/6)	17%	(2/12)	0%	(0/14)	0%	(0/17)	12%	(9/77)
	2	17%	(2/12)	6%	(1/16)	0%	(0/7)	8%	(1/12)	0%	(0/14)	0%	(0/16)	5%	(4/77)
3	1	25%	(6/24)	35%	(7/20)	22%	(6/27)	11%	(3/27)	36%	(9/25)	14%	(3/21)	24%	(34/144)
	2	38%	(9/24)	38%	(8/21)	27%	(6/22)	35%	(7/20)	19%	(5/27)	15%	(3/20)	28%	(38/134)
	Totals	15%	(22/148)	18%	(26/143)	14%	(18/128)	11%	(17/149)	11%	(17/152)	7%	(10/148)	13%	(110/868)

## Data Availability

Data is contained within the article.

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
