# Peer review of "Genetic Listeria monocytogenes Types in the Pork Processing Plant Environment: From Occasional Introduction to Plausible Persistence in Harborage Sites"

_pathogens, 2021, doi:10.3390/pathogens10060717_

Round 1

Reviewer 1 Report

Dear Authors, Dear Editor,

submitted manuscript describes an interesting study on genetic diversity of Listeria monocytogenes in pork cutting plant environment, and reveals the role of occasional introduction and persistence of the pathogen in production facilities. The study design in simple and clearly described. Alongside, during review following questions, remarks and comments have to be addressed by authors to improve the manuscript:

1) Page 2, lines 66-68: please clarify or include information in the table, i.e. number of positive samples in different zones represents sampling during production or after C&D in each plant!

2) Page 5 and Page 6 includes Figure 2, which are two separate tables! Please correct that and make relevant references in the text!

3) Page 4, lines 98-99, Table 1- please, add clarifications in the table column replace "Production" with "During production" to clarify sampling site and time!

4) Page 4, line 101- clarify LM isolate collection sites or plants or origin of isolates!

5) Page 4, line101- clarify what is meant by "serotyping PCR analysis"

6) How do you explain- why IIc serotype of one CC is predominant?

7) Can you describe more in details how long (time) persistence of LM was observed during this study... weeks, months?

8) M&M Chapter 4.1. please make separate subchapters on selection and description of cutting plants AND environmental sampling to have a better overview!

9) Why have you chosen to use sponge sticks instead of sponges to have more efficient sampling on surfaces with pressure to obtain good samples on surfaces where biofilms could exist?

10)What was a LM confirmation procedure?, if you have used ISO 11290-1 standard!

11) There is missing more detailed information on pork cutting plants:

  • how old were production facilities, last renovations, hygiene design of the equipment, where there any differences observed between plants, evaluation/ assessment of GHP's;
  • where there any connection with slaughterhouses or cutting plants were fully isolated? etc.

Overall, one of the study tasks was to highlight/ present well-designed extensive sampling programmes combined with genetic characterization! Can you describe and promote some cost-efficient approach when designing those plans for SME pork cutting plants?!

Although, study looks simple, clear and interesting for meat businesses, scientific quality of the manuscript could be improved by answering above mentioned questions as well as formulating more sound conclusions.

Wish you success in improving the manuscript!

Sincerely,

Reviewer #

Author Response

Dear Reviewer,

Thank you very much for reading and reviewing this manuscript. I have addressed each of your concerns as outlined in the revisions manuscript (Please see the attachment). All suggestions from both reviewers have been taken into account in hopes of meeting your expectations.

Thank you for your consideration and all the best in this difficult period.

Sincerely,

Niels Demaître

Reviewer 2 Report

Revision of manuscript 1203288

Genetic Listeria monocytogenes types in the pork cutting plant environment: from occasional introduction to plausible persistence in harborage sites

by  Demaître  et al.

General comments

The manuscript reports the detection of L. monocytogenes in three pork cutting plants, with the aim to detect harborage sites and persistent clones.

The topic is interesting and worthy of investigation.

However, a deeper genotyping (i.e. WGS) and longer sampling intervals would be needed to define repeatedly isolated strains as “persistent clones”.

I suggest prudent use of “CC” and “persistent”, preferably opting to more appropriate definitions.

English should be revised by a native speaker in the entire manuscript.

Specific comments

Abstract

L. 20-22. Do clonal complexed derive from PFGE or WGS analysis?

Results

L. 104-106. The correlation between CC and pulsotypes should be better explained, here or in the Methods section.

About results in paragraph 2.2.1 An additional table or figure could clarify the isolation frequency of each CC separately in each plant, indicating if the isolation occurred in the first or in the second sampling. Otherwise, frequencies can also be added in Fig. 2.

Discussion

L. 291-295. I agree with authors when they say “The question arises to what extent isolates should be characterized to determine persistence and provide support at the cutting plant level.” However, it is true that “WGS-based typing, which has higher discriminative power than PFGE analysis, should further divide the genetic groups at the plant level.” For this reason, and especially considering the higher discriminatory power of WGS with respect to PFGE, I think that it is not completely correct to use the term CC to define PFGE-related strains found in this study. I suggest to avoid the term CC substituting it with “pulsotypes”.

Materials and methods

L. 310-311. Please indicate the modern and the old cutting plant with the letters A, B and C, referring to the same letters used in subsequent line 322.

L. 312-313. Are two samplings within 9 months sufficient to define strains as “persistent”? Is the sampling period enough wide? In light of this, L. 279-281 appears to be contradictory.

Bibliographic references are needed. See also Guidi et al. 2021 Microorganisms 2021, 9, 376 as an example.

L. 315. Why 4-6 weeks? Is there any recommendation or guideline? Which was the criteria to select this interval?

L. 318-320. Figure 1 does not distinguish among the three cutting plants, thus the citation of Figure 1 in this phrase is not completely correct. Instead, it is appropriate in lines 344.

L. 327. Can authors provide information about detergents and disinfectants used for the C&D actions?

L. 357-359. Swab samples homogenized in which solution? 20 ml of 2x Fraser added to what?

L. 372. If isolates were confirmed by multiplex PCR and serotyping, why do you define them “suspect”? Maybe it should be as follows: Suspected isolates were confirmed by multiplex PCR and serotyping, then confirmed strains were PFGE typed.

L. 377-378. I wonder if it is completely correct to deduce the CC from the pulsotype. The two methods have very different discriminatory powers (see also comments to the Discussion section) and are not comparable. Moreover, cited references appear to be old (2016 and 2018).

Author Response

(The authors gave the same response as above.)

Round 2

Reviewer 1 Report

Comments and questions raised were addressed properly!